# BPE Stays on SCRIPT: Structured Encoding
# for Robust Multilingual Pretokenization

**Sander Land** [1]   **Catherine Arnett** [2]

## Abstract

Byte Pair Encoding (BPE) tokenizers, widely used in Large Language Models, face challenges in multilingual settings, including penalization of non-Western scripts and the creation of tokens with partial UTF-8 sequences. Pretokenization, often reliant on complex regular expressions, can also introduce fragility and unexpected edge cases. We propose SCRIPT (Script Category Representation in PreTokenization), a novel encoding scheme that bypasses UTF-8 byte conversion by using initial tokens based on Unicode script and category properties. This approach enables a simple, rule-based pretokenization strategy that respects script boundaries, offering a robust alternative to pretokenization strategies based on regular expressions. We also introduce and validate a constrained BPE merging strategy that enforces character integrity, applicable to both SCRIPT-BPE and byte-based BPE. Our experiments demonstrate that SCRIPT-BPE achieves competitive compression while eliminating encoding-based penalties for non-Latin-script languages.

  github.com/sanderland/script_bpe

## 1. Introduction

Text representation for language models involves a fundamental mismatch between legacy encoding systems and modern NLP practices. This mismatch is evident during pretokenization. Byte Pair Encoding (BPE, Sennrich et al., 2016), the dominant tokenization method, typically operates after an initial pretokenization step, during which the input text is split into smaller 'pretokens'. Then BPE merges these into more meaningful tokens. However, pretokenization commonly relies on manually crafted regular expressions, which can be difficult to create and interpret. Reliance on

these patterns often leads to unexpected edge cases and suboptimal segmentation, particularly in diverse multilingual contexts (Velayuthan & Sarveswaran, 2025; Land, 2024).

Beyond these pretokenization difficulties, the choice of character representation itself introduces additional biases. BPE typically operates either on UTF-8 bytes (byte-level BPE) or Unicode characters (character-level BPE). However, both approaches present fundamental challenges. In byte-level BPE, the variable encoding length implicitly penalizes non-Latin scripts ('byte premium effects', Arnett et al., 2024). For some scripts, characters are represented with a single byte (e.g., Latin), while other scripts have characters with two or three bytes per character (e.g., Greek and Chinese, respectively). This is illustrated in Figure 1A. In addition, the representation of characters is disconnected from their meaning, with similar characters having very different representations, or very different characters sharing a common prefix purely due to Unicode's historical assignment order, as illustrated in Figure 1B.

Byte-level BPE tokenizers can also learn merges that cross character boundaries, resulting in tokens representing a mix of full and partial characters. Although tokens representing part of a *single* character's UTF-8 encoding are necessary to form three- and four-byte characters, those representing a mix of full and partial characters lack semantic meaning, risk invalid encodings, and can become under-trained (Land & Bartolo, 2024). Due to the greedy nature of BPE, an early merge crossing character boundaries can lead to a cascade of merges that create ever more such tokens (Figure 1C). The GPT-4o tokenizer has 874 such tokens, e.g. `<0x95>\n\n`.

Although applying BPE directly to Unicode characters, e.g. SentencePiece (Kudo & Richardson, 2018), avoids initial UTF-8 conversion, it introduces a different challenge: Unicode uses approximately 150,000 codepoints to represent a wide range of characters. This forces tokenizers to initially select a manageable subset of Unicode codepoints using coverage parameters, then to fall back to UTF-8 bytes or out-of-vocabulary tokens for unselected characters.

To address these fundamental limitations of both UTF-8 and Unicode approaches, we introduce SCRIPT-BPE, a novel encoding scheme built on Unicode script and category

[1]Cohere [2]EleutherAI. Correspondence to: Sander Land <sander@cohere.com>.

*Proceedings of the ICML 2025 Tokenization Workshop (TokShop)*, Vancouver, Canada. PMLR 267, 2025. Copyright 2025 by the author(s).

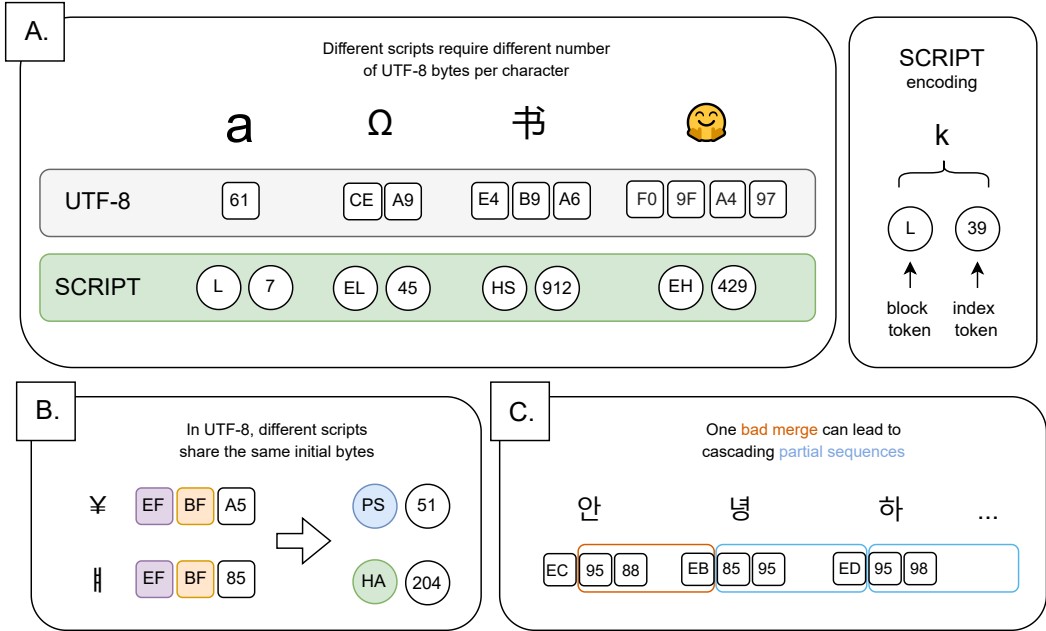

*Figure 1.* A: Illustration of variable encoding length in UTF-8 encoding compared to consistent encoding length for SCRIPT encoding. B: In UTF-8, the same initial byte sequences may be used for characters in different scripts, however SCRIPT encoding uses consistent block tokens to represent characters from the same script block. C: Byte-based BPE can allow merges that create partial UTF-8 sequences, which then cascade through the rest of the sequence.

properties. Our encoding uses two initial tokens to represent each character, eliminating crosslinguistic bias due to byte premium effects. This encoding naturally leads to a simple rule-based pretokenization algorithm that groups characters by script properties, offering a robust alternative to regular expressions. Additionally, we propose a *constrained BPE merging* strategy. We show that this method drastically reduces partial character sequences in both SCRIPT- and byte-based BPE.

## 2. Related Work

**Pretokenization.** Recent work shows that pretokenization can have a greater impact on downstream performance than more salient parameters like vocabulary size (Wegmann et al., 2025). Despite this, it is a relatively under-studied area within tokenizer research and a small number of approaches are dominant, each with their own weaknesses.

Many pretokenizers are complex regular expressions, mostly variants of the GPT-2 pretokenizer. Generally, they have rules to split based on English contractions, whitespaces, and digits. Pretokenizing English contractions like *'m* (e.g. *I'm*), *'s* (e.g. *woman's*), *'d* (e.g. *I'd*), *etc.* primarily benefits English, by providing specialized handling that preserves word structure. These rules, however, can harm tokenization for other languages (Arnett, 2024). For example, the follow-ing words would be targeted by the same regular expression, but instead of preserving word structure, it would split the word at an unnatural point: *s'mhath* ('(it's) good'; Scottish Gaelic), *m'sit* ('all', Mi'kmaq), *n'di* ('eat'; Fulfulde), and *'dan* ('son'; Hausa).

Whitespace pretokenization is pervasive as a pretokenization approach—even for pretokenizers that don't rely on regular expressions. Recent work shows that removing whitespace pretokenization or allowing merges across whitespace boundaries leads to improved compression, lower latency, and higher throughput (Liu et al., 2025; Schmidt et al., 2025). This early work questions whether whitespace-based pretokenization is an essential component for tokenization.

Digit pretokenization (Schmidt et al., 2024) involves splitting all digits into individual pretokens. The GPT regular expression splits strings of digits into groups of up to three digits. Single-digit tokenization is associated with better performance in arithmetic tasks when compared to regular-expression-based or whole-number tokenization (Liu & Low, 2023; Dagan et al., 2024; Singh & Strouse, 2024).

Regular-expression-based pretokenizers have also been shown to be fragile, even for English data. For instance, Schmidt et al. (2025) show that the original GPT pretokenizer doesn't account for different straight versus curly apostrophes (e.g. isn't and isn't, respectively) and propose a

more robust regular-expression-based pretokenizer to handle these cases.

**Language Parity.** As illustrated in Figure 1A, the variable encoding length of UTF-8 leads to byte premiums for different languages (Arnett et al., 2024). Recent efforts have sought to address these tokenization challenges. MYTE (Limisiewicz et al., 2024) proposes morphology-driven byte encodings for fairer representations, which lead to more equitable tokenization across languages. This approach requires morphologically annotated data, however, so is limited in its ability to scale to new languages.

Lee et al. (2025) propose *jamo*-level over syllable-level tokenization for Korean, which offers benefits for low-resource machine translation. Again, this approach is limited in its language coverage, as it only works for languages that use Hangul. However, language-specific solutions like this highlight the benefits of using linguistically informed units for tokenization.

Velayuthan & Sarveswaran (2025) highlight how common pretokenizers split diacritics from their corresponding characters for scripts used for Tamil, Sinhala, and Hindi. The authors propose grapheme-level as opposed to unicode-codepoint-level to keep characters and their diacritics together and avoid edge cases for the regular expression. This leads to improved compression for those languages.

## 3. SCRIPT Encoding

Our novel encoding scheme, SCRIPT (Script Category Representation in PreTokenization), maps each Unicode character to a pair: a *block token* and an *index token*. This mapping is derived from two Unicode properties:

- **Unicode Script:** The writing system to which a character belongs (e.g., Latin, Cyrillic, Han).

- **Unicode Supercategory:** We define five supercategories by grouping standard Unicode general categories: Letters & Marks (LM), Punctuation & Symbols (PS), Numbers (N), Separators (Z), and Other (C). Precise definitions and exceptions are in Appendix A.

Each unique Script-Supercategory pair defines a potential character block. As can be seen in Table 1, some of these blocks contain a very large number of characters. To manage vocabulary size, we split any Script block larger than a predefined threshold into multiple sub-blocks. We set this threshold to the size of the 'Latin LM' block. This results in 468 *block tokens* (each representing a specific script-supercategory-sub-block combination), and 1448 *index tokens* (identifying a character within a block).

By representing each Unicode character as a unique pair

of a block token and an index token, we both reduce the worst-case encoding length of a character from four byte-tokens in byte-based BPE to two tokens. At the same time, this approach also provides more linguistically meaningful representations.

### 3.1. SCRIPT pretokenization

Structured SCRIPT encoding enables a simple rule-based pretokenization strategy, avoiding the need for complex regular expressions. In this approach, we form initial groups of consecutive characters that share the same Unicode script and supercategory. These initial groups are then refined to form the final pretokens by merging groups involving single spaces or with an 'Inherited' script property. Full details of this pretokenization algorithm are provided in Appendix B.

### 3.2. Constrained BPE Merges

Standard Byte Pair Encoding (BPE) greedily merges the most frequent adjacent tokens. In our SCRIPT-BPE scheme—which represents each character as a pair of a block token and an index token—an unconstrained BPE could, for instance, merge an index token $i_1$ from one character with the block token $b_2$ of the subsequent character. Such cross-character merges would lead to tokens similar to partial UTF-8 sequence tokens (Land & Bartolo, 2024) and introduce complexities during inference, potentially requiring models to learn intricate dependencies between partial character representations, similar to the case presented in Figure 1C.

To avoid these problematic merges, we experiment with a *constrained merge* strategy. Specifically for SCRIPT-BPE, constrained BPE merges are only allowed between tokens that already represent one or more full characters,

*Table 1.* Largest character counts per Script-Supercategory block. We split the large highlighted blocks into sub-blocks.

| Script | Supercat. | Size |
|---|---|---|
| Han | LM | 98,687 |
| Hangul | LM | 11,677 |
| Common | PS | 7,195 |
| Tangut | LM | 6,914 |
| Egyptian Hieroglyphs | LM | 5,089 |
| Latin | LM | 1,448 |
| Arabic | LM | 1,253 |
| Yi | LM | 1,165 |
| Cuneiform | LM | 1,118 |
| Common | LM | 1,049 |
| Canadian Aboriginal | LM | 723 |
| Inherited | LM | 655 |
| Bamum | LM | 641 |
| Common | N | 586 |
| Anatolian Hieroglyphs | LM | 583 |

or between a single block token and a single index token (thereby forming a complete character).

We also apply a similar technique to standard byte-level BPE. We introduce a constraint where merges are allowed *within* the byte sequence of a single character (processed strictly left-to-right), or *between* sequences that each represent complete characters. It disallows combining a space character (a full character) with only the first byte of a subsequent multi-byte character, which is a common early merge in unconstrained BPE on multi-byte scripts.

# 4. Results

We train and evaluate tokenizers in monolingual and multilingual settings. Monolingual tokenizers are trained with a vocabulary size of 64,000 on 300 MB subsets sourced from Chang et al. (2024). We train monolingual tokenizers for 12 languages.[1] Multilingual tokenizers are trained to 256,000 merges on a 35 GB subsample of CulturaX (Nguyen et al., 2023) to provide broad multilingual coverage. We also create a 136 GB validation set.[2]

In both settings, we compare UTF-8 byte-based BPE with SCRIPT-BPE. The byte-based tokenizers use `tiktoken`'s regular expression-based pretokenizers (OpenAI, 2024). We try both `cl100k` (GPT-4) and `o200k` (GPT-4o) variants. For SCRIPT-BPE tokenizers, we test both our proposed rule-based and regular expression pretokenization.

## 4.1. Constrained merges

As shown in Table 2, we find that constraining BPE merges to respect character boundaries is beneficial for all encodings and datasets. Not only does it eliminate tokens with partial character sequences, but it also almost universally improves compression.

Byte-based BPE with o200k regular expression shows a particular outlier on the Thai dataset, with 42,831 tokens representing a mix of full and partial characters, as a result of several early merges between common characters and two leading UTF-8 bytes `<0xE0><0xB8>`, causing a cascade of merges with partial characters. As differences in compression are generally small, we present only the constrained versions in all subsequent results.

## 4.2. Implementation and Performance

Our tokenizer training experiments are conducted using a custom Python implementation. Although constraining merges introduces additional boundary checks during the

---

[1] The languages in our sample are Japanese, Chinese, Thai, Punjabi, Hindi, Korean, Russian, Arabic, Hebrew, Vietnamese, German, and English chosen to represent a diverse range of scripts.

[2] Datasets linked in repository

BPE process, it also reduces the size of internal data structures used in training by limiting the search space. Table 3 presents the training time performance. For SCRIPT-based approaches, constraining merges reduces training time compared to unconstrained versions (cf. ✓ vs. ✗). For byte-based tokenizers, training time increases slightly with constrained merges, potentially due to the more complex character boundary checks required for UTF-8 encoding compared to SCRIPT. In practice, for a moderately parallel setup with 16 CPUs, all tokenizers train in approximately one hour for 256,000 merges on the multilingual dataset, ensuring that training time is not a bottleneck.

## 4.3. Compression

Table 4 presents compression rates for our multilingual tokenizer across various languages, showing both initial character encoding costs and final compression ratios after BPE merges. Here we calculate compression as the number of tokens per character over the validation set. Choice of pretokenization significantly influences these final compression ratios, in line with findings from Wegmann et al. (2025). Notably, for Thai, Hindi, and Punjabi, the pattern used by 'cl100k' splits words at diacritic marks, severely worsening compression. In contrast, the SCRIPT-based pretokenization generally achieves robust compression across diverse scripts. However, it can lag behind the more complex 'o200k' pretokenization pattern in specific cases such as Chinese and Thai. For Chinese, this difference may be

*Table 2.* Results for constraining merges to form full Unicode characters first (✓) versus the normal non-constrained approach (✗). Tokens/Char shows mean compression ratio on the training corpora for the monolingual tokenizers in Tokens/Unicode character, and #Partial Char. shows mean count of tokens with partial character sequences.

| | Tokens/Char | | #Partial Char. | |
|---|---|---|---|---|
| **Constrained Merges:** | ✗ | ✓ | ✗ | ✓ |
| Bytes + cl100k regex | 0.333 | 0.332 | 1678 | 209 |
| Bytes + o200k regex | 0.282 | 0.280 | 5193 | 183 |
| SCRIPT (rule-based) | 0.293 | 0.289 | 6048 | 0 |
| SCRIPT + o200k regex | 0.284 | 0.280 | 8062 | 0 |

*Table 3.* Training time in hours for the multilingual tokenizer with 256k merges with different pretokenizers, not including pretokenization and initialization.

| **Compute:** | **1 CPU** | | **16 CPUs** | |
|---|---|---|---|---|
| **Constrained Merges:** | ✗ | ✓ | ✗ | ✓ |
| Bytes + cl100k regex | 5.2 | 6.7 | 1.2 | 1.3 |
| Bytes + o200k regex | 5.8 | 8.0 | 1.2 | 1.3 |
| SCRIPT (rule-based) | 4.4 | 4.1 | 1.0 | 0.9 |
| SCRIPT + o200k regex | 6.1 | 4.5 | 1.1 | 1.0 |

*Table 4.* We report performance of the multilingual tokenizer on its training set, and both monolingual and multilingual validation sets. Initial tokens/character refers to the number of tokens used to encode the dataset before merges. Compression results are shown for both UTF-8 byte encoding, and SCRIPT encoding, where 'rule-based' refers to the SCRIPT encoding's novel pretokenization strategy. Lower values are better. Values $> 5\%$ , $> 10\%$ , and $> 20\%$ worse than the best in their row are highlighted.

| | Init. Tokens/Char | | Final Tokens/Character | | | |
| | | | Bytes | | SCRIPT | |
| Language (Script) | Bytes | SCRIPT | cl100k | o200k | rule-based | o200k |
|---|---|---|---|---|---|---|
| Japanese (Japanese) | 2.74 | 2 | 0.5249 | 0.5268 | 0.5393 | 0.5267 |
| Chinese (Han) | 2.69 | — | 0.6244 | 0.6260 | 0.6537 | 0.6259 |
| Thai (Thai) | 2.68 | — | 0.4257 | 0.3160 | 0.3362 | 0.3159 |
| Punjabi (Gurmukhi) | 2.54 | — | 0.6053 | 0.4982 | 0.4962 | 0.4979 |
| Hindi (Devanagari) | 2.51 | — | 0.5057 | 0.3246 | 0.3233 | 0.3245 |
| Korean (Hangul) | 2.33 | — | 0.5808 | 0.5827 | 0.5817 | 0.5824 |
| Russian (Cyrillic) | 1.81 | — | 0.2314 | 0.2321 | 0.2321 | 0.2320 |
| Arabic (Arabic) | 1.79 | — | 0.2979 | 0.2976 | 0.2963 | 0.2975 |
| Hebrew (Hebrew) | 1.77 | — | 0.4146 | 0.4163 | 0.4157 | 0.4161 |
| Vietnamese (Latin) | 1.32 | — | 0.2688 | 0.2692 | 0.2687 | 0.2692 |
| German (Latin) | 1.02 | — | 0.2128 | 0.2133 | 0.2138 | 0.2132 |
| English (Latin) | 1.01 | — | 0.2152 | 0.2150 | 0.2179 | 0.2150 |
| Mean monolingual | 2.02 | — | 0.4090 | 0.3765 | 0.3812 | 0.3764 |
| CulturaX Training Data | 1.24 | — | 0.2559 | 0.2537 | 0.2587 | 0.2536 |
| CulturaX Validation Data | 1.23 | — | 0.2526 | 0.2509 | 0.2560 | 0.2508 |

attributed to mixed Chinese/Latin phrases often found in web data (e.g. spam or advertisements) and the prevalence of non-standard use of spaces.

The compression results for tokenizers trained on individual monolingual datasets closely matched those observed with the multilingual tokenizer. For completeness, we provide these results in Appendix C.

## 5. Discussion and Conclusion

Our novel encoding scheme shows promising results for more fair and robust text representation. The SCRIPT-BPE approach, combining a novel SCRIPT encoding with rule-based pretokenization and constrained BPE merging, achieves competitive compression while mitigating several common pitfalls of traditional tokenizers. Notably, the simple constraint of enforcing character boundaries during BPE merging universally eliminated tokens representing a mix of full and partial characters and generally improved compression across different base encodings. Given its compatibility with all encoding approaches, we recommend its adoption, particularly for massively multilingual tokenizers

This preliminary evaluation focused primarily on compression; however, this metric alone does not necessarily guarantee better downstream model performance (Schmidt et al., 2024). For instance, not using any pretokenization will achieve high raw compression[3] but has also been shown to

---
[3]Around 12% higher on average in our experiments.

have poor downstream performance (*ibid*).

In future work, we aim to train language models using both the SCRIPT and the constrained merging strategy and evaluate their effects on model performance. This is essential for understanding their true impact on downstream task performance, generalization, and fairness at scale. As our proposed method is computationally efficient, it does not represent a barrier to scaling these methods to training large language models with SCRIPT-BPE. There is also room to further refine the SCRIPT encoding and pretokenization itself. For example, refining the handling of digits and punctuation could help bridge performance gaps observed with specialized regular expressions like o200k. Another promising direction enabled by our encoding scheme is the possibility of developing modular, script-specific tokenizers, which can be combined as needed for the intended downstream purpose. The SCRIPT framework is also inherently extensible as future additions to the Unicode standard can be incorporated by defining new block tokens without altering the core logic. While SCRIPT provides an alternative to regex-based pretokenizers, it remains compatible with regex rules when beneficial, opening opportunities to explore which rule combinations would yield additional benefits.

Overall, SCRIPT-BPE provides a robust and extensible framework for developing more robust and equitable tokenization systems, with significant potential for improving multilingual language models.

## Impact Statement

We hope this paper contributes to improved language parity in machine learning. By reducing encoding biases against non-Western scripts, our work may help create more equitable language models that better serve diverse linguistic communities.

## Acknowledgments

We thank Matthias Gallé, Felipe Cruz-Salinas, and James Owers-Bardsley for their valuable feedback on the manuscript.

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

## A. Script and Supercategory definition

For the purpose of our pretokenization scheme, we define several *supercategories* for characters. This involves grouping existing Unicode general categories and manually reassigning specific characters from their default Unicode categories or script properties. These adjustments are made to better align character classifications with their practical usage in our context.

The Unicode general categories for Letters (L*) and Marks (M*) are treated as a single supercategory 'LM'. This is a natural grouping, as marks (e.g., diacritics, accents) are typically attached to or modify letters.

The Unicode general categories for Punctuation (P*) and Symbols (S*) are combined into another supercategory 'PS'. The boundary between punctuation and symbols can be ambiguous, and characters from both categories often serve similar roles, especially in source code. For example, the common programming operators `->` and `!=` consist of a character from the Unicode Punctuation category followed by one from the Symbol category.

In addition, we manually re-assign the following characters:

- The newline (`\n`, U+000A) and tab (`\t`, U+0009) characters are re-assigned to the Separator category (Z). Unicode classifies these as Other/Control characters for historical reasons. This change allows us to treat newlines and tabs the same as other whitespace characters during pretokenization.

- The Katakana-Hiragana prolonged sound marks (U+30FC and U+FF70) are re-assigned to the 'Inherited' script. These marks are originally classified under the 'Common' script due to their use with both Katakana and Hiragana. By reassigning them to the 'Inherited' script, we allow them to be grouped with either Katakana or Hiragana characters during pretokenization.

- The 'Tatweel' mark (U+0640) used in Arabic text justification is reassigned from Common to the Arabic script.

Characters in the Unassigned (Cn), Private Use Area (Co), and Surrogate (Cs) Unicode categories are excluded from encoding and are effectively filtered out. These code points do not have a defined representation in Unicode (Cn), are intended for custom use by software (Co), or are reserved for UTF-16 encoding mechanics (Cs). Such characters often originate from artifacts or proprietary formatting and are not meaningful or generalizable for language modeling. To ensure fairness, the same filtering is applied consistently to all baseline methods during evaluation.

Figure 2 shows the distribution of the block sizes after these reassignments.

## B. SCRIPT-based pretokenization

Our rule-based pretokenization strategy creates pretokens with a consistent script and category. Beyond handling leading spaces (a common practice), a few additional rules are needed to capture edge cases in Unicode. The specific steps for our pretokenization algorithm are the following:

- **Initial Script-Based Grouping:** Group consecutive characters that share the same Unicode script and supercategory. For this step, sub-blocks within larger script/supercategory combinations are ignored.

- **Space Merging:** If a group consists of a single space character, it may be merged with the following group. This occurs when the following group is either an LM supercategory from scripts which use whitespace to separate words[4] or a 'Common PS' group.

- **Inherited Script Merging:** If a group's script is 'Inherited' (characters that depend on the preceding character, like combining diacritics), the group is merged with the preceding group, and any following groups that share the initial group's script and supercategory. For example, a sequence of groups such as (Arabic LM, Inherited LM, Arabic LM, Inherited LM, Arabic LM) will be merged into a single group.

- **Hiragana-Han Merging:** Additionally, sequences of Han and Hiragana characters are merged into a single group, preventing splits within Japanese words and grammatical constructions that mix Kanji and Hiragana.

---

[4]Our current list consists of: Latin, Arabic, Devanagari, Hangul, Ethiopic, Cyrillic, Greek, Hebrew, Bengali, Syriac, Oriya, Tamil, Telugu, Gurmukhi, Gujarati, Sinhala, Malayalam, Armenian, Kannada, Georgian

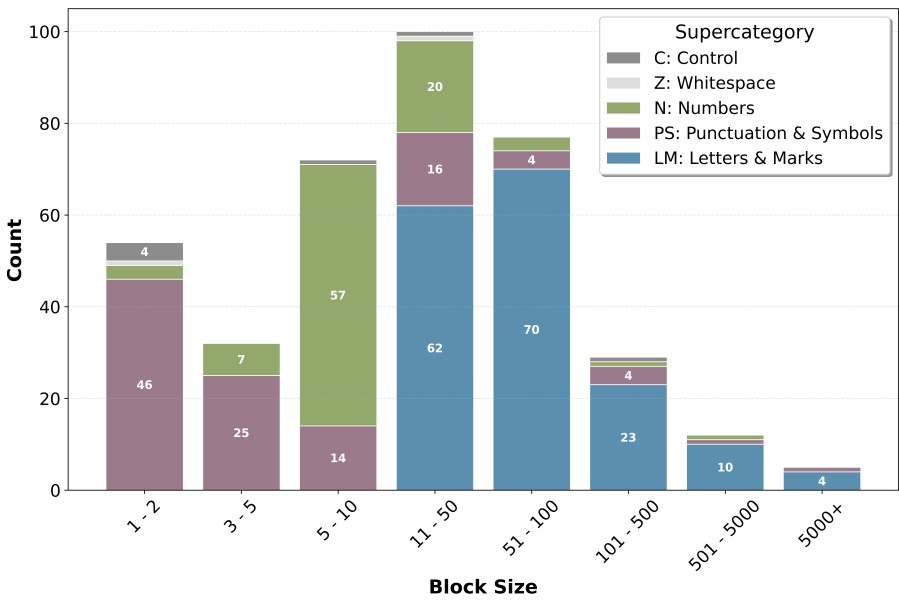

*Figure 2.* Distribution of block sizes in SCRIPT encoding, before splitting large blocks into sub-blocks.

## C. Monolingual tokenizer performance

Compression performance for monolingual tokenizers on their training data (Table 5) shows similar patterns to validation performance of the larger multilingual tokenizer on monolingual datasets shown in Table 4.

*Table 5.* Final compression ratios (Tokens/Unicode character after BPE merges) across languages and tokenizer types for monolingual models. Lower compression values are better. Values > 5% , > 10% , and > 20% worse than the best in their row are highlighted.

|  | Tokens/Character | | | |
| --- | --- | --- | --- | --- |
|  | **Byte cl100k** | **Byte o200k** | **SCRIPT rule-based** | **SCRIPT o200k** |
| Japanese | 0.4196 | 0.4197 | 0.4422 | 0.4194 |
| Chinese | 0.5136 | 0.5137 | 0.5587 | 0.5134 |
| Thai | 0.3216 | 0.2048 | 0.2364 | 0.2047 |
| Punjabi | 0.4961 | 0.2398 | 0.2398 | 0.2394 |
| Hindi | 0.4835 | 0.2387 | 0.2387 | 0.2387 |
| Korean | 0.3945 | 0.3945 | 0.3973 | 0.3942 |
| Russian | 0.2115 | 0.2115 | 0.2123 | 0.2115 |
| Arabic | 0.2325 | 0.2299 | 0.2298 | 0.2299 |
| Hebrew | 0.2439 | 0.2426 | 0.2446 | 0.2427 |
| Vietnamese | 0.2552 | 0.2553 | 0.2553 | 0.2553 |
| German | 0.2007 | 0.2008 | 0.2019 | 0.2008 |
| English | 0.2118 | 0.2114 | 0.2145 | 0.2114 |
| Mean | 0.3320 | 0.2802 | 0.2893 | 0.2801 |

