# OpenReview forum: "BPE Stays on SCRIPT: Structured Encoding for Robust Multilingual Pretokenization"
_ICML.cc/2025/Workshop/TokShop — TokShop_

### Official Review · Reviewer_3Y8P · 2025-06-06
**A promising method for linguistically motivated pretokenization**

**Rating:** 8
**Confidence:** 4

**Review:**

The authors present their novel, linguistically motivated, pretokenization scheme aimed at tackling the common problems emerging during pretokenization of the non-Latin scripts using the popular tokenizers. They clearly describe their encoding algorithm and demonstrate its compression potential on both monolingual and multilingual corpora covering wide variety of different writing scripts. Their comparison with the current state-of-the-art pretokenization shows that their method can achieve similar token/character compression rates to the SotA without sacrificing computation time (the algorithm performed slightly worse on glyph languages such as Chinese and Japanese, however, they provide a hypothesis behind the performance drop).

The authors mention that their submission is a preliminary study not showing the effect of the proposed encoding when applied to LLM training, however, it is understandable given the format of short-paper. The authors also do not forget to mention possible directions for further development of the encoding method.

I have only one question related the encoding (it is possible that I have overlooked it in the paper). How does the encoding (byte mapping) method handle future expansion of the utf-8 character set, e.g. by a possible expansion to scripts not yet covered by the standard? Do the authors have a suggestion for handling such case?

---

### Official Review · Reviewer_ZrYb · 2025-06-08
**A good step forward**

**Rating:** 8
**Confidence:** 5

**Review:**

This is a lovely short paper with a promising future direction.

$\textbf{Summary}$

This paper introduces an interesting solution for pre-tokenization used in Byte Pair Encoding (BPE) tokenizers, called SCRIPT (Script Category Representation in Pre-Tokenization). This is a prevailing issue, particularly for non-English languages. The authors present a novel encoding schema that addresses both the existing pre-tokenization challenges and the byte premium effect. Their schema encodes each character using two tokens, thereby mitigating the byte premium issue.

They conduct intrinsic evaluations on both monolingual and multilingual datasets. Additionally, the authors propose a constrained BPE merging strategy that enforces character integrity. This strategy can be applied to both their novel SCRIPT-BPE and standard byte-based BPE tokenizers. The results demonstrate that this is a promising solution; however, future work is needed to evaluate how SCRIPT-BPE performs during large language model (LLM) training and on downstream tasks.

$\textbf{Strengths}$
1. This paper introduces a novel approach to a persisting problem in NLP.
2. The initial results indicate a promising direction for the SCRIPT methodology.
3. Honestly, I love this approach. I have always wanted to see a new encoding schema that brings all languages to a level playing field from an encoding perspective. The only issue I had with introducing a new scheme was the need for a bridge between Unicode (the current standard for dataset encoding) and the proposed scheme. Since this method operates directly on Unicode, the mapping is fairly straightforward, allowing it to be adopted in tokenizer training and LLM training.
4. A diverse set of languages has been considered in this evaluation, demonstrating that the schema is language-agnostic.

$\textbf{Weakness}$
1. The languages considered are mainly high-resource or mid-resource. A study involving known low-resource languages would have been helpful.
2. Although it is not within the scope of the paper, it would have been useful to see how this tokenizer performs on an actual LLM.

$\textbf{Suggestion to improve paper structure}$
1. A table describing the languages used, their language families, and their degree of morphological richness would give the reader a clearer perspective on the diversity of the languages considered.

---

### Official Review · Reviewer_UKhD · 2025-06-08
**SCRIPT-BPE - Better treatment of non-Latin scripts in tokenization**

**Rating:** 8
**Confidence:** 4

**Review:**

This paper introduces SCRIPT-BPE, a new tokenization technique design to address limitations/biases of BPE tokenizers used in LLMs, especially against non-Latin script languages. The proposed method tries to create a more equitable multilingual text representation by eliminating the "byte premium" that penalizes non-Latin scripts. Key ideas introduced in the paper include
1. SCRIPT Encoding scheme which maps each Unicode character to a pair: a block token and an index token, reducing worst-case character encoding from 4 byte-tokens to 2 tokens and creating consistent encoding length across all scripts, eliminating bias against non-Latin languages
2. Rule-based pre-tokenization strategy which replaces complex regular expressions with simple rules based on Unicode script properties there by appropriately handling edge cases like inherited scripts and space merging through straightforward rules
3. Constrained BPE Merges which eliminates problematic partial UTF-8 sequence tokens, and prevents merges that cross character boundaries

**Strengths:**
- The SCRIPT encoding scheme proposed is technically sound in addressing the root-cause of multilingual bias in tokenization rather than applying hacky band-aid solutions.
- Usage of script and category unicode properties to create a more meaningful character representations is grounded in fundamental principles of linguistics.
- The simplicity of rule based pre-tokenization makes it easily interpretable and maintainable compared to complex regular expression based approached
- Sound methodological rigor is evident from the paper where the specific issues of current tokenizers are clearly articulated, previous related work was quoted and consistent evaluation metrics and datasets used.
- Evaluation of the proposed approach was observed to be comprehensive with 12 diverse languages with different scripts considered, and  compared against 2 baselines cl100k and o200k.
- The idea introduced shows promising results across monolingual and multilingual datasets, demonstrating good training times for real-world practical applications

**Weaknesses:**
- One key limitation of the presented work is that all results are based on tokenizer metrics without training actual language models. This lack of downstream task performance leaves a gap before practical adoption (rightly acknowledged by the authors as essential future work). It would have been more comprehensive if inference time considerations were also presented.
- Inclusion of failure cases or problematic character sequences would greatly improve the result analysis, strengthening the case for the proposed idea

Overall, this paper makes a solid technical contribution to the tokenization literature, and provides a foundation for more equitable multilingual language models.

---

### Decision · Program_Chairs · 2025-06-10

Accept